# Fixed-Time-Convergent Sliding Mode Control with Sliding Mode Observer for PMSM Speed Regulation

**DOI:** 10.3390/s24051561

**Published:** 2024-02-28

**Authors:** Xin Zhang, Hongwen Li, Meng Shao

**Affiliations:** 1Changchun Institute of Optics, Fine Mechanics and Physics, Chinese Academy of Sciences, Changchun 130033, China; zhangxin171@mails.ucas.ac.cn (X.Z.); shaomeng@ciomp.ac.cn (M.S.); 2University of Chinese Academy of Sciences, Beijing 100049, China

**Keywords:** permanent magnet synchronous motor (PMSM), sliding mode control (SMC), sliding mode observer, fixed-time control, speed control

## Abstract

This paper focuses on the speed control of a permanent magnet synchronous motor (PMSM) for electric drives with model uncertainties and external disturbances. Conventional sliding mode control (CSMC) can only converge asymptotically in the infinite domain and will cause unacceptable sliding mode chattering. To improve the performance of the PMSM speed loop in terms of response speed, tracking accuracy, and robustness, a hybrid control strategy for a fixed-time-convergent sliding mode controller (FSMC) with a fixed-time-convergent sliding mode observer (FSMO) is proposed for PMSM speed regulation using the fixed-time control theory. Firstly, the FSMC is proposed to improve the convergence speed and robustness of the speed loop, which can converge to the origin within a fixed time independent of the initial conditions. Then, the FSMO is used as a compensator to further enhance the robustness of the speed loop and attenuate sliding mode chattering. Finally, simulation and experimental results show that the proposed method can effectively improve the dynamic performance and robustness of the PMSM speed control system.

## 1. Introduction

The permanent magnet synchronous motor (PMSM) has gained widespread adoption in various domains, including robotics, electric vehicles, and large-aperture telescopes, owing to its inherent benefits such as high power density, a favorable torque ratio, and a broad speed range [1,2,3]. The Field-Oriented Control (FOC) strategy is a prevalent choice in industrial applications for PMSM control due to its rapid and fully decoupled regulation of torque and flux [4]. In an FOC-based PMSM drive system, a commonly employed configuration involves a cascade control structure with two inner current control loops and an outer speed control loop [5]. The efficiency and reliable operation of the motor are directly influenced by the outer speed loop. However, mode uncertainties and external load disturbances contribute to a reduction in the tracking accuracy and stability of the PMSM speed control system [6]. Simultaneously, traditional control methods such as the Proportional–Integral (PI) controller are vulnerable to external disturbances and internal parameter mismatches, leading to deviations from the expected target.

Therefore, numerous studies devoted to developing appropriate control techniques in the field of PMSMs have been carried out [7]. Several nonlinear control methods have been proposed and implemented, e.g., active disturbance rejection control [8,9], predictive current control [10], internal model control [11], and sliding mode control (SMC) [12]. Among these modern control methods, SMC is a popular and effective strategy for nonlinear systems with disturbances due to its benefits like quick response, small overshoot, and strong disturbance rejection ability [13]. In the literature [14], a conventional sliding mode controller with an adaptive law was designed to improve the dynamic response speed and robustness of the PMSM drive system. Huang et al. proposed a traditional adaptive sliding mode controller with a series-structure resonant controller for the PMSM speed loop to effectively attenuate the disturbances, including model uncertainties, external load, and torque ripple [15]. The setting time is an important performance index of SMC. However, the sliding surface of conventional SMC (CSMC) is a linear combination of the system states, and the settling time of CSMC is infinite, theoretically [16]. In order to improve the convergence speed of CSMC, the fixed-time control theory is being studied by more and more researchers because its setting time is bounded by a fixed value which is independent of the initial system conditions [17]. In the literature [18], fixed-time, nonsingular terminal SMC based on the fixed-time stable theory is proposed for second-order nonlinear systems. Cao et al. has designed a reinforcement-learning-based, fixed-time, nonsingular, fast terminal sliding mode for trajectory tracking control of uncertain robotic manipulators with input saturation [19]. In the literature [20], Ni et al. proposed a fast, fixed-time, nonsingular terminal sliding mode control method and applied it to design the energy storage device controller and SVC controller for chaos suppression in power systems. Although, the above fixed-time terminal SMC methods are effective in different application areas, little research has studied the application of fixed-time SMC in PMSM systems. Moreover, the terminal SMC strategy, whose sliding mode surface contains the differential term of the error, is not suitable for the first-order PMSM speed control system. Therefore, a new fixed-time-convergent SMC (FSMC) scheme will be a promising and effective method to improve the response speed and anti-disturbance property for PMSM speed regulation.

In formulating an SMC controller, it is essential to carefully determine an appropriate switching gain. When the speed loop experiences significant disruptions, such as parameter mismatches and external loads, designing a substantial switching gain for the SMC speed controller becomes crucial to ensure system stability and anti-disturbance capability [21]. Nevertheless, an elevated switching gain may result in a discontinuous control signal and severe sliding mode chattering characterized by high-frequency oscillations [22]. Furthermore, in practical PMSM systems, obtaining precise information about the maximum system disturbance proves challenging due to inherent uncertainties. Consequently, to mitigate the effects of a high switching gain and eliminate the associated high-frequency sliding mode chattering, the unknown nonlinear lumped disturbance requires estimation and compensation within the FSMC controller. Addressing these challenges involves leveraging a disturbance observer to enhance system control performance, given its proficiency in handling external disturbances [23]. Numerous studies have demonstrated that disturbance observers utilizing the sliding mode technique can effectively achieve the desired estimation performance. The sliding mode observer (SMO) is particularly advantageous due to its ease of application and robustness [24], and it has been successfully applied to various different systems [25,26,27,28]. In prior works [29,30], external disturbances were observed using the SMO, followed by the implementation of feedback compensation based on the observed values where the switching gain only needed to surpass the upper bound of the disturbance compensation error. Therefore, a fixed-time-convergent SMO is proposed to attenuate the sliding mode chattering and further enhance the anti-disturbance capability of the system.

According to the previous discussion, this paper proposes a fixed-time-convergent sliding mode control method with a fixed-time-convergent sliding mode observer (FSMC-FSMO) for PMSM speed regulation. The main contributions and novelty of this article are summarized as follows:An FSMC is proposed for the PMSM speed loop to achieve the fixed-time convergence property, thereby improving the convergence speed and robustness of the CSMC;An FSMO is designed to observe the unknown nonlinear lumped disturbance including mode uncertainties and external load torque. Meanwhile, the observed lumped disturbance is used to compensate the FSMC to further improve the robustness of the PMSM speed control system and effectively attenuate high-frequency sliding mode chattering;The stability and fixed-time convergence property of the proposed method are proofed by the Lyapunov method, and the feasibility and effectiveness are verified by the simulation and experimental results.

This paper is organized as follows. In Section 2, the mathematical model of the PMSM with the model uncertainties and external disturbances is introduced, and the main theoretical foundations are given. In Section 3, the CSMC and proposed FSMC-FSMO are designed, and the stability and fixed-time convergence property of the proposed method are discussed. In Section 4, simulation and experiments are performed to demonstrate the effectiveness of the proposed control strategy. Section 5 concludes this paper.

## 2. Mathematical Model and Theoretical Foundations

### 2.1. Mathematical Model of the Permanent Magnet Synchronous Motor

For the surface-mounted PMSM, ignoring the effects of hysteresis loss, eddy current, and core saturation, the voltage equation of the PMSM in the synchronous rotating coordinate system can be described as:(1)$$\left\{\begin{array}{c} u_{d}=R_{s}i_{d}+L_{d}\frac{di_{d}}{dt}-n_{p}\omega L_{q}i_{q}\\ u_{q}=R_{s}i_{q}+L_{q}\frac{di_{q}}{dt}+n_{p}\omega \left(L_{d}i_{d}+\psi_{f}\right) \end{array}\right.$$
where $i_{d}$ and $u_{d}$ represent the *d*-axis stator current and voltage, respectively; $i_{q}$ and $u_{q}$ represent the q-axis stator current and voltage, respectively; $R_{s}$ is the stator resistance; $L_{d}$ and $L_{q}$ are the *d*-axis inductance and *q*-axis inductance of the stator winding, respectively; $n_{p}$ is the number of pole pairs; ω represents the rotor mechanical angular velocity; $\psi_{f}$ represents the rotor flux linkage.

The dynamic equation of PMSM can be expressed as:(2)$$J\dot{\omega }=T_{e}-B\omega -T_{l}$$
where *J* is the moment of inertia; *B* is the viscous friction coefficient; $T_{e}$ is the electromagnetic torque; and $T_{l}$ is the load torque.

The electromagnetic torque can be presented as:(3)$$T_{e}=\frac{3}{2}n_{p}i_{q}\left[i_{d}\left(L_{d}-L_{q}\right)+\psi_{f}\right]=\frac{3}{2}n_{p}\psi_{f}i_{q}=K_{t}i_{q}$$
where $K_{t}$ is the torque coefficient.

For PMSM drive systems with model uncertainties and external disturbances, the moment of inertia, viscous friction coefficient, and rotor flux linkage will change with changes in operating environment and load. The change of these parameters can be expressed as:(4)$$\left\{\begin{array}{c} \Delta J=J-J_{0}\\ \Delta B=B-B_{0}\\ \Delta \psi_{f}=\psi_{f}-\psi_{f0}\\ \Delta K_{t}=K_{t}-K_{t0} \end{array}\right.$$
where $J_{0}$, $B_{0}$, $\psi_{f0}$, and $K_{t0}$ are the nominal parameters; ΔJ, ΔB, $\Delta \psi_{f}$, and $\Delta K_{t}$ represent the parameter variations.

Considering the parameter mismatch, the dynamic equation of the PMSM can be rewritten as:(5)$$\left(J_{0}+\Delta J\right)\dot{\omega }=\left(K_{t0}+\Delta K_{t}\right)i_{q}-\left(B_{0}+\Delta B\right)\omega -T_{l}$$

Then, Equation (Equation 5) can be further rewritten as:(6)$$\dot{\omega }=\frac{K_{t0}}{J_{0}}i_{q}-\frac{B_{0}}{J_{0}}\omega -d$$
where $d=\Delta J\dot{\omega }-\Delta K_{t}i_{q}+\Delta B\omega +T_{l}$ is the unknown total disturbance including parameter mismatch and external load torque.

### 2.2. A New Fixed-Time Stable System

**Definition 1** ([17])**.**
*Consider the system with the following differential equation:*
(7)$$\dot{x}\left(t\right)=f\left(x\left(t\right)\right),x\left(0\right)=x_{0}$$
*The origin of *(Equation 7)* is called a fixed-time stable equilibrium point if it is globally finite-time stable with a bounded settling time function $T(x_{0})$, i.e., $\exists T_{\max}>0$ such that $T\left(x_{0}\right)<T_{\max}$.*


Then, according to Definition 1, a fixed-time stable system is given in Lemma 1.

**Lemma 1.** 
*Consider the system with the following differential equation:*

(8)
$$\dot{y}=-\alpha {\mathrm{sgn}}^{p}\left(y\right)-\beta {\mathrm{sgn}}^{q}\left(y\right),y\left(0\right)=y_{0}$$

*where α>0, β>0, 0<p<1, q>1, and $sgn^{*}\left(y\right)={\left|y\right|}^{*}\cdot sign\left(y\right)$, where sign(y) is the sign function. Then, system *(Equation 8)* is a fixed-time stable system whose settling time is upper bounded by:*

(9)
$$T<\frac{1}{\alpha }\frac{1}{1-p}+\frac{1}{\beta }\frac{1}{q-1}$$



**Proof of Lemma 1.** Define a positive semidefinite function $V\left(y\right)=y^{2}$. Differentiating V(y) along system (Equation 8) yields:
(10)$$\begin{array}{cc} \dot{V}\left(y\right) & =2y[-\alpha {\mathrm{sgn}}^{p}\left(y\right)-\beta {\mathrm{sgn}}^{q}\left(y\right)]\\ & =-2\alpha {\left(y^{2}\right)}^{\frac{1+p}{2}}-2\beta {\left(y^{2}\right)}^{\frac{1+q}{2}}\\ & =-2\left(\alpha +\beta V^{\frac{1+q}{2}-\frac{1+p}{2}}\right)V^{\frac{1+p}{2}}\leq 0 \end{array}$$Assuming that V(y)≠0, Equation (Equation 10) can be further expressed as:
(11)$$\begin{array}{c} \frac{1}{V^{\frac{1+p}{2}}}\frac{dV}{dt}=-2\left(\alpha +\beta V^{\frac{1+q}{2}-\frac{1+p}{2}}\right)\\ \Rightarrow \frac{1}{1-p}\frac{dV^{\frac{1-p}{2}}}{dt}=-\left(\alpha +\beta V^{\frac{1+q}{2}-\frac{1+p}{2}}\right) \end{array}$$Defining $\bar{V}=V^{\frac{1-p}{2}}$, Equation (Equation 11) can be written as:
(12)$$\begin{array}{c} \;\;\;\frac{1}{1-p}\frac{d\bar{V}}{dt}=-\left(\alpha +\beta {\bar{V}}^{1+\frac{q-1}{1-p}}\right)\\ \Rightarrow \frac{1}{\alpha +\beta {\bar{V}}^{1+\frac{q-1}{1-p}}}d\bar{V}=\left(p-1\right)dt \end{array}$$Integrating both sides of Equation (Equation 12) can obtain:
(13)$$\begin{array}{c} \;\;\int_{{\bar{V}}_{0}}^{0}\frac{1}{\alpha +\beta {\bar{V}}^{1+\frac{q-1}{1-p}}}d\bar{V}=\int_{0}^{T({\bar{V}}_{0})}(p-1)dt\\ \Rightarrow T\left({\bar{V}}_{0}\right)=\frac{1}{1-p}\int_{0}^{{\bar{V}}_{0}}\frac{1}{\alpha +\beta {\bar{V}}^{1+\frac{q-1}{1-p}}}d\bar{V} \end{array}$$Then, the upper bound of the settling time can be estimated as:
(14)$$\begin{array}{cc} \lim_{{\bar{V}}_{0}\to \infty }T\left({\bar{V}}_{0}\right) & =\frac{1}{1-p}\lim_{{\bar{V}}_{0}\to \infty }\int_{0}^{{\bar{V}}_{0}}\frac{1}{\alpha +\beta {\bar{V}}^{1+\frac{q-1}{1-p}}}d\bar{V}\\ & =\frac{1}{1-p}\left(\int_{0}^{1}\frac{1}{\alpha +\beta {\bar{V}}^{1+\frac{q-1}{1-p}}}d\bar{V}+\int_{1}^{\infty }\frac{1}{\alpha +\beta {\bar{V}}^{1+\frac{q-1}{1-p}}}d\bar{V}\right)\\ & <\frac{1}{1-p}\left(\int_{0}^{1}\frac{1}{\alpha }d\bar{V}+\int_{1}^{\infty }\frac{1}{\beta {\bar{V}}^{1+\frac{q-1}{1-p}}}d\bar{V}\right)\\ & =\frac{1}{1-p}\left(\frac{1}{\alpha }+\frac{1}{\beta }\frac{1-p}{q-1}\right)\\ & =\frac{1}{\alpha }\frac{1}{1-p}+\frac{1}{\beta }\frac{1}{q-1} \end{array}$$Due to $\bar{V}=V^{\frac{1-p}{2}}=y^{1-p}$, $\bar{V}\to \infty$ as y→∞, and $\bar{V}\to 0$ as y→0, and if and only if y=0, $\bar{V}=0$. Thus, it can be concluded that the settling time $T(y_{0})$ can also be bounded by $\frac{1}{\alpha }\frac{1}{1-p}+\frac{1}{\beta }\frac{1}{q-1}$. The proof is completed. □

## 3. Design of Speed Controller Based on the Fixed-Time-Convergent Sliding Mode Control with a Fixed-Time-Convergent Sliding Mode Observer Method

In order to achieve fast, robust, and smooth PMSM speed control performance, this section designs an FSMC-FSMO control strategy. Firstly, the CSMC is designed to compare with the proposed method. Then, the FSMC is designed to improve the convergence speed of the CSMC by applying fixed-time control theory to the CSMC, which can converge to the origin within a fixed time independent of the initial system error. Next, the FSMO is designed to further compensate the unknown disturbance to enhance the robustness of the system and attenuate sliding mode chattering. Finally, the stability and fixed-time convergence property of the FSMC-FSMO are proofed using the Lyapunov method.

### 3.1. Design of the Conventional Sliding Mode Control

For the dynamic system of the PMSM (Equation 6), to remove the steady-state error and ensure the precision of speed control, a integral sliding surface can be designed as:(15)$$s_{c}=e+k_{c1}\int_{0}^{t}ed\tau$$
where $k_{c1}>0$ is the integral coefficient of the CSMC, and $e=\omega_{ref}-\omega$ is the speed error where $\omega_{ref}$ is the speed reference.

The derivative of the sliding surface is called the reaching law, which has various forms, summarized in [31]. In this paper, the exponential reaching law is adopted to design the CSMC, which is given as follows:(16)$${\dot{s}}_{c}=-k_{c2}s_{c}-\mu_{c}sign\left(s_{c}\right)$$
where $k_{c2}>0$ and $\mu_{c}>0$ are the exponential coefficient and switching gain of the CSMC.

Then, taking the derivative of the sliding surface (Equation 15) on both sides yields:(17)$$\begin{array}{cc} {\dot{s}}_{c} & =\dot{e}+k_{c1}e\\ \; & ={\dot{\omega }}_{ref}-\frac{K_{t0}}{J_{0}}i_{q}+\frac{B_{0}}{J_{0}}\omega +d+k_{c1}e \end{array}$$

Due to *d* being the unknown total disturbance, substituting (Equation 16) into (Equation 17) yields:(18)$$i_{q,ref}=\frac{J_{0}}{K_{t0}}\left[{\dot{\omega }}_{ref}+\frac{B_{0}}{J_{0}}\omega +k_{c1}e+k_{c2}s_{c}+\mu_{c}sign\left(s_{c}\right)\right]$$
where $\mu_{c}>\left|d\right|$, and $i_{q,ref}$ is the *q*-axis current reference.

### 3.2. Design of the Fixed-Time-Convergent Sliding Mode Controller

For the dynamic system of the PMSM (Equation 6), to improve the response speed and achieve the fixed-time convergence property of speed control, a fixed-time-convergent integral sliding surface can be designed as:(19)$$s=e+k_{1}\int_{0}^{t}[\lambda_{1}{\mathrm{sgn}}^{p_{1}}\left(e\right)+{\mathrm{sgn}}^{q_{1}}\left(e\right)]d\tau$$
where $k_{1}>0$ is the integral coefficient of the FSMC, $\lambda_{1}>0$, $0<p_{1}<1$, and $q_{1}>1$.

Usually, the basic sliding motion of SMC can be divided into two steps, as shown in Figure 1. The first step is to force the system trajectory to move from a random initial state towards the sliding surface. This step is defined as the reaching process. The second step is to slide along the sliding mode towards the origin after the system trajectory reaches the sliding surface. This step is defined as the sliding process. The fixed-time-convergent integral sliding surface designed in Equation (Equation 19) can only ensure that the system converges within a fixed time during the sliding process. Thus, a fixed-time-convergent sliding mode reaching law is designed as:(20)$$\dot{s}=-k_{2}\left[\lambda_{2}{\mathrm{sgn}}^{p_{2}}\left(s\right)+{\mathrm{sgn}}^{q_{2}}\left(s\right)\right]-\mu sign\left(s\right)$$
where $k_{2}>0$, $\lambda_{2}>0$, $0<p_{2}<1$, $q_{2}>1$, and μ>0 is the switching gain of the FSMC.

Then, taking the derivative of the sliding surface (Equation 19) on both sides yields:(21)$$\begin{array}{cc} \dot{s} & =\dot{e}+k_{1}\left[\lambda_{1}{\mathrm{sgn}}^{p_{1}}\left(e\right)+{\mathrm{sgn}}^{q_{1}}\left(e\right)\right]\\ \; & ={\dot{\omega }}_{ref}-\frac{K_{t0}}{J_{0}}i_{q}+\frac{B_{0}}{J_{0}}\omega +d+k_{1}\left[\lambda_{1}{\mathrm{sgn}}^{p_{1}}\left(e\right)+{\mathrm{sgn}}^{q_{1}}\left(e\right)\right] \end{array}$$

Considering *d* as a disturbance term, substituting (Equation 20) into (Equation 21) yields:(22)$$\begin{array}{cc} i_{q,ref}=\frac{J_{0}}{K_{t0}} & \{{\dot{\omega }}_{ref}+\frac{B_{0}}{J_{0}}\omega +k_{1}\left[\lambda_{1}{\mathrm{sgn}}^{p_{1}}\left(e\right)+{\mathrm{sgn}}^{q_{1}}\left(e\right)\right]\\ \; & +k_{2}\left[\lambda_{2}{\mathrm{sgn}}^{p_{2}}\left(s\right)+{\mathrm{sgn}}^{q_{2}}\left(s\right)\right]+\mu sign\left(s\right)\} \end{array}$$
where μ>|d|.

**Remark 1.** 
*Since d is the unknown total disturbance of the system, it is difficult to obtain the value of d. Therefore, it is difficult to give a suitable value for μ. If the value of μ is too much larger than |d|, the system will produce severe high-frequency sliding mode chattering; if the value of μ is too much smaller than |d|, the system will have poor robustness. Therefore, it is necessary to compensate for the unknown disturbance d.*


### 3.3. Design of the Fixed-Time-Convergent Sliding Mode Observer

In order to further improve the robustness of the FSMC and attenuate sliding mode chattering, an FSMO is designed to compensate for the unknown disturbance *d* in the PMSM dynamic system.

Firstly, according to the literature [14], in a practical PMSM drive system, the system disturbances vary very slowly compared with the system state in every sampling period of the speed loop. Thus, the derivative of the total unknown disturbance *d* in Equation (Equation 6) with respect to time *t* can be regarded as $\dot{d}\left(t\right)=0$. Thus, the dynamic system of the PMSM can be expressed as:(23)$$\left\{\begin{array}{cc} \dot{\omega } & =\frac{K_{t0}}{J_{0}}i_{q}-\frac{B_{0}}{J_{0}}\omega -d\\ \dot{d} & =0 \end{array}\right.$$

Then, considering the mechanical rotor angular speed ω and the unknown total disturbance *d* as the state variables, the *q*-axis current $i_{q}$ as the control input, and ω as the system output, the FSMO can be designed as:(24)$$\left\{\begin{array}{cc} \dot{\hat{\omega }} & =\frac{K_{t0}}{J_{0}}i_{q}-\frac{B_{0}}{J_{0}}\hat{\omega }-\hat{d}+f\left(\hat{e}\right)\\ \dot{\hat{d}} & =\rho f(\hat{e}) \end{array}\right.$$
where $\hat{\omega }$ and $\hat{d}$ are the observations of the mechanical rotor angular speed ω and the unknown total disturbance *d*, respectively; $\hat{e}=\omega -\hat{\omega }$ is the speed observation error; $f(\hat{e})$ is the sliding mode function; and ρ<0 is the observer gain.

According to Equations (Equation 23) and (Equation 24), the observation error differential equation can be obtained as follows:(25)$$\left\{\begin{array}{c} \dot{\hat{e}}=-\frac{B_{0}}{J_{0}}\hat{e}-{\hat{e}}_{d}-f\left(\hat{e}\right)\\ {\dot{\hat{e}}}_{d}=-\rho f\left(\hat{e}\right) \end{array}\right.$$
where ${\hat{e}}_{d}=d-\hat{d}$ is the disturbance observation error.

Then, define the fixed-time-convergent sliding surface of the FSMO as:(26)$$\hat{s}=\hat{e}+k_{o1}\int_{0}^{t}[\lambda_{o1}{\mathrm{sgn}}^{p_{o1}}\left(\hat{e}\right)+{\mathrm{sgn}}^{q_{o1}}\left(\hat{e}\right)]d\tau$$
where $k_{o1}>0$ is the integral coefficient of the FSMO, $\lambda_{o1}>0$, $0<p_{o1}<1$, and $q_{o1}>1$.

Define the fixed-time-convergent sliding mode reaching law of the FSMO as:(27)$$\dot{\hat{s}}=-k_{o2}\left[\lambda_{o2}{\mathrm{sgn}}^{p_{o2}}\left(\hat{s}\right)+{\mathrm{sgn}}^{q_{o2}}\left(\hat{s}\right)\right]-\mu_{o}sign\left(\hat{s}\right)$$
where $k_{o2}>0$, $\lambda_{o2}>0$, $0<p_{o2}<1$, $q_{o2}>1$, and $\mu_{o}>0$ is the switching gain of the FSMO.

Considering ${\hat{e}}_{d}$ as the disturbance term, substituting Equations (Equation 25) and (Equation 26) into Equation (Equation 27) yields:(28)$$f\left(\hat{e}\right)=-\frac{B_{0}}{J_{0}}\hat{e}+k_{o1}\left[\lambda_{o1}{\mathrm{sgn}}^{p_{o1}}\left(\hat{e}\right)+{\mathrm{sgn}}^{q_{o1}}\left(\hat{e}\right)\right]+k_{o2}\left[\lambda_{o2}{\mathrm{sgn}}^{p_{o2}}\left(\hat{s}\right)+{\mathrm{sgn}}^{q_{o2}}\left(\hat{s}\right)\right]+\mu_{o}sign\left(\hat{s}\right)$$
where $\mu_{o}>\left|{\hat{e}}_{d}\right|$.

**Theorem 1.** 
*The FSMO of the PMSM dynamic system is fixed-time stable, which can converge to the origin within a fixed time independent of the initial conditions of the observer. The settling time is upper bounded by:*

(29)
$$\left\{\begin{array}{c} T_{o}=T_{o1}+T_{o2}\\ T_{o1}<\frac{1}{k_{o1}\lambda_{o1}}\frac{1}{1-p_{o1}}+\frac{1}{k_{o1}}\frac{1}{q_{o1}-1}\\ T_{o2}<\frac{1}{k_{o2}\lambda_{o2}}\frac{1}{1-p_{o2}}+\frac{1}{k_{o2}}\frac{1}{q_{o2}-1} \end{array}\right.$$

*where $T_{o}$ is the total settling time, $T_{o1}$ is the settling time of the sliding process, and $T_{o2}$ is the settling time of the reaching process.*


**Proof of Theorem 1.** Defining a positive semidefinite Lyapunov function $L_{o}={\hat{s}}^{2}$ and taking the derivative of $L_{o}$ can give:
(30)$$\begin{array}{cc} {\dot{L}}_{o} & =2\hat{s}\dot{\hat{s}}\\ & =2\hat{s}\left\{\dot{\hat{e}}+k_{o1}\left[\lambda_{o1}\mathrm{sg}n^{p_{o1}}\left(\hat{e}\right)+\mathrm{sg}n^{q_{o1}}\left(\hat{e}\right)\right]\right\}\\ & =2\hat{s}\left\{-\frac{B_{0}}{J_{0}}\hat{e}-{\hat{e}}_{d}-f\left(\hat{e}\right)+k_{o1}\left[\lambda_{o1}\mathrm{sg}n^{p_{o1}}\left(\hat{e}\right)+\mathrm{sg}n^{q_{o1}}\left(\hat{e}\right)\right]\right\}\\ & =2\hat{s}\left\{-k_{o2}\left[\lambda_{o2}\mathrm{sg}n^{p_{o2}}\left(\hat{s}\right)+\mathrm{sg}n^{q_{o2}}\left(\hat{s}\right)\right]-\mu_{o}sign\left(\hat{s}\right)-{\hat{e}}_{d}\right\}\\ & \leq 2\hat{s}\left[-\lambda_{o2}\mathrm{sg}n^{p_{o2}}\left(\hat{s}\right)-k_{o2}\mathrm{sg}n^{q_{o2}}\left(\hat{s}\right)\right]\leq 0 \end{array}$$It can be concluded that $L_{o}\geq 0$, ${\dot{L}}_{o}\leq 0$, and $\hat{s}$ is bounded. When $L_{o}=0$, $\hat{s}=0$. According to LaSalle’s invariance theorem [32], the FSMO has global asymptotic stability. Then, for the inequality (Equation 30), when the equal sign holds, $L_{o}$ has the slowest convergence speed. Thereby, according to Lemma 1, the settling time $T_{o2}$ of the reaching process ($\hat{s}:{\hat{s}}_{0}\to 0$) can be bounded by $T_{o2}\leq \frac{1}{k_{o2}\lambda_{o2}}\frac{1}{1-p_{o2}}+\frac{1}{k_{o2}}\frac{1}{q_{o2}-1}$.When the FSMO reaches the sliding surface $\hat{s}=0$, taking the derivative on both sides of Equation (Equation 26) yields:
(31)$$\dot{\hat{e}}=-k_{o1}\lambda_{o1}\mathrm{sg}n^{p_{o1}}\left(\hat{e}\right)-k_{o1}\mathrm{sg}n^{q_{o1}}\left(\hat{e}\right)$$Thus, according to Lemma 1, the convergence time of the speed observation error $\hat{e}$ can be bounded by $T_{o1}<\frac{1}{k_{o1}\lambda_{o1}}\frac{1}{1-p_{o1}}+\frac{1}{k_{o1}}\frac{1}{q_{o1}-1}$.Based on the above analysis, the total settling time of the FSMO is $T_{o}=T_{o1}+T_{o2}$. The proof is completed. □

### 3.4. The Proposed Fixed-Time-Convergent Sliding Mode Control with a Fixed-Time-Convergent Sliding Mode Observer

To further enhance the robustness of the FSMC and attenuate sliding mode chattering, compensating the total disturbance to the FSMC observed by the FSMO constitutes the proposed FSMC-FSMO strategy. According to Equation (Equation 22), the control law of the FSMC compensated by the FSMO can be presented as:(32)$$\begin{array}{cc} i_{q,ref}=\frac{J_{0}}{K_{t0}} & \{{\dot{\omega }}_{ref}+\frac{B_{0}}{J_{0}}\omega +\hat{d}+k_{1}\left[\lambda_{1}{\mathrm{sgn}}^{p_{1}}\left(e\right)+{\mathrm{sgn}}^{q_{1}}\left(e\right)\right]\\ \; & +k_{2}\left[\lambda_{2}{\mathrm{sgn}}^{p_{2}}\left(s\right)+{\mathrm{sgn}}^{q_{2}}\left(s\right)\right]+\mu sign\left(s\right)\} \end{array}$$
where $\mu >|{\hat{e}}_{d}|$.

**Theorem 2.** 
*The dynamic system *(Equation 6)* of the PMSM is made fixed-time stable using the fixed-time sliding control law *(Equation 32)*, which can converge to the origin within a fixed time independent of the initial system conditions. The settling time of the system is upper bounded by:*

(33)
$$\left\{\begin{array}{c} T=T_{1}+T_{2}\\ T_{1}<\frac{1}{k_{1}\lambda_{1}}\frac{1}{1-p_{1}}+\frac{1}{k_{1}}\frac{1}{q_{1}-1}\\ T_{2}<\frac{1}{k_{2}\lambda_{2}}\frac{1}{1-p_{2}}+\frac{1}{k_{2}}\frac{1}{q_{2}-1} \end{array}\right.$$

*where T is the total settling time, $T_{1}$ is the settling time of the sliding process, and $T_{2}$ is the settling time of the reaching process.*


**Proof of Theorem 2.** Defining a positive semidefinite Lyapunov function $L=s^{2}$ and taking the derivative of *L* yields:
(34)$$\begin{array}{cc} \dot{L} & =2s\dot{s}\\ & =2s\{\dot{e}+k_{1}\left[\lambda_{1}\mathrm{sg}n^{p_{1}}\left(e\right)+\mathrm{sg}n^{q_{1}}\left(e\right)\right]\}\\ & =2s\{{\dot{\omega }}_{ref}-\frac{K_{t0}}{J_{0}}i_{q}+\frac{B_{0}}{J_{0}}\omega +d+k_{1}\left[\lambda_{1}\mathrm{sg}n^{p_{1}}\left(e\right)+\mathrm{sg}n^{q_{1}}\left(e\right)\right]\} \end{array}$$Substituting the fixed-time sliding control law (Equation 32) into (Equation 34) yields:
(35)$$\begin{array}{cc} \dot{L} & =2s\{d-\hat{d}-k_{2}\left[\lambda_{2}\mathrm{sg}n^{p_{2}}\left(s\right)+\mathrm{sg}n^{q_{2}}\left(s\right)\right]-\mu sign\left(s\right)\}\\ & \leq 2s[-k_{2}\lambda_{2}\mathrm{sg}n^{p_{2}}\left(s\right)-k_{2}\mathrm{sg}n^{q_{2}}\left(s\right)]\leq 0 \end{array}$$It can be concluded that L≥0, $\dot{L}\leq 0$, and *s* is bounded. When L=0, s=0. According to LaSalle’s invariance theorem [31], the system has global asymptotic stability. Then, for the inequality (Equation 35), when the equal sign holds, *L* has the slowest convergence speed. Therefore, according to Lemma 1, the settling time $T_{2}$ of the reaching process ($s:s_{0}\to 0$) can be bounded by $T_{2}\leq \frac{1}{k_{2}\lambda_{2}}\frac{1}{1-p_{2}}+\frac{1}{k_{2}}\frac{1}{q_{2}-1}$.When the system reaches the sliding surface s=0, taking the derivative on both sides of Equation (Equation 19) yields:
(36)$$\dot{e}=-k_{1}\lambda_{1}\mathrm{sg}n^{p_{1}}\left(e\right)-k_{1}\mathrm{sg}n^{q_{1}}\left(e\right)$$Then, according to Lemma 1, the convergence time of the speed error *e* can be bounded by $T_{1}<\frac{1}{k_{1}\lambda_{1}}\frac{1}{1-p_{1}}+\frac{1}{k_{1}}\frac{1}{q_{1}-1}$.Based on the above analysis, the total settling time of the system is $T=T_{1}+T_{2}$. The proof is completed. □

**Remark 2.** 
*In the control law of the FSMC *(Equation 22)*, the switching gain μ>|d| will cause severe sliding mode chattering when the total disturbance d is large. On the contrary, the switching gain in the control law of the FSMC-FSMO is $\mu >|\hat{e}|$, which is much smaller than |d| and does not change with d. Therefore, the robustness of the speed control system has been enhanced by the proposed FSMC-FSMO method, and the sliding mode chattering is effectively attenuated.*


Finally, the FOC-based PMSM control scheme with the proposed method is shown in Figure 2. The current loop applies two identical PI controllers. The speed loop applies the proposed FSMC-FSMO controller. The proposed scheme improves the response speed and robustness of the PMSM speed loop. Moreover, it can effectively suppress the sliding mode chattering phenomenon.

## 4. Simulation and Experimental Results

In order to verify the feasibility and effectiveness of the proposed method, comparative simulation and experimental results of the CSMC, FSMC, and FSMC-FSMO methods are given based on the structural diagram of the PMSM drive system with the FSMC-FSMO scheme (Figure 2). The nominal parameters of the PMSM used for simulations and experiments are shown in Table 1. The controller parameters of the CSMC are configured as $k_{c1}=k_{c2}=5$ and $\mu_{c}=0.05$. For a fair comparison, the parameters $k_{1}$, $k_{2}$, and μ of the FSMC are also configured as $k_{1}=k_{2}=5$ and μ=0.05. The other parameters of the FSMC are selected as $\lambda_{1}=\lambda_{2}=1$, $p_{1}=p_{2}=0.8$, and $q_{1}=q_{2}=1.2$. The parameters of the FSMO are configured as $k_{o1}=k_{o2}=10$, $\mu_{o}=0.05$, $\lambda_{o1}=\lambda_{o2}=1$, $p_{o1}=p_{o2}=0.8$, $q_{o1}=q_{o2}=1.2$, and ρ=−10.

### 4.1. Simulation Results and Analysis

#### 4.1.1. The Fixed-Time Convergence Property Analysis

In order to analyze the fixed-time convergence property of the FSMC, Figure 3 shows the comparative simulation results of the CSMC and FSMC methods under different initial conditions when d=0. It can be seen from Figure 3a that the convergence time of the speed error with the CSMC and FSMC methods under $\omega_{ref}=50$ rpm is 0.72 s and 1.33 s, respectively. Meanwhile, in order to highlight the ability of the proposed FSMC method to converge in a fixed time independent of the initial conditions, Figure 3b presents the simulation results of the CSMC and FSMC under a very large reference speed of 50,000 rpm for comparison with the simulation results under the reference speed of 50 rpm. It can be seen from Figure 3b that the convergence time of the speed error with the CSMC and FSMC methods under $\omega_{ref}=50,000$ rpm is 0.86 s and 2.55 s, respectively. Therefore, it can concluded that the proposed FSMC has faster convergence speed than the CESC under different initial conditions. Comparing Figure 3a,b, it can be seen that the convergence time of the FSMC increases by 0.14 s when the initial speed reference changes from 50 rpm to 50,000 rpm. On the contrary, the convergence time of the CSMC increases by 1.22 s, which is much larger than that of the FSMC. Thereby, the fixed-time convergence property of the FSMC is verified.

#### 4.1.2. Speed Response Performance and Robustness Analysis

To analyze the speed response performance and robustness of the proposed FSMC-FSMO method, Figure 4 and Figure 5 respectively show the comparative simulation results of the CSMC, FSMC, and FSMC-FSMO methods in tracking step speed signal and sine speed signal under load conditions. Firstly, Figure 4 shows the comparative simulation results of the three methods in tracking step speed signal $\omega_{ref}=100$ rpm under the conditions of adding rated load at 5 s and removing rated load at 10 s. It can be seen from Figure 4 that the settling time of the CSMC, FSMC, and FSMC-FSMO without load is 1.41 s, 0.72 s, and 0.72 s, respectively. Thereby, the FSMC has a faster speed response than the CSMC, and the FSMO will not affect the speed response performance of the FSMC. When a rated load is suddenly added at 5 s, the speed drop and settling time of the CSMC are 18.7 rpm and 1.62 s, respectively. With the FSMC, the speed drop and settling time are reduced to 9.3 rpm and 0.97 s, respectively. Using the FSMC-FSMO method, the speed drop is further reduced to 6.9 rpm. When a rated load is suddenly removed at 10 s, the speed rise and settling time of the CSMC are 18.7 rpm and 1.62 s, respectively. With the FSMC, the speed rise and settling time are reduced to 9.3 rpm and 0.97 s, respectively. Using the FSMC-FSMO method, the speed rise is further reduced to 6.9 rpm. Therefore, it can be concluded that the FSMC has faster convergence speed and stronger robustness than the CSMC, and the FSMO can further improve the robustness of the FSMC to the unknown disturbance. Moreover, it can be seen from Figure 4d that the FSMO can effectively observe the lumped disturbance.

Then, Figure 5 shows the comparative simulation results of the three methods in tracking sinusoidal speed signal $\omega_{ref}=60+30sin\left(2t\right)$ rpm under the conditions of adding rated load at 5 s and removing rated load at 10 s. It can be seen from Figure 5 that the performance of the three methods when tracking a sinusoidal speed reference is similar to that when tracking a step speed reference. As can be seen from Figure 5, the settling time of the speed error of CSMC is 1.35 s, which is shorter than the settling time of tracking the 100 rpm step speed signal. In contrast, the settling time of the speed error of the FSMC and FSMC-FSMO is the same as the settling time of tracking the 100 rpm step speed signal, which is achieved by the fixed-time convergence property of the FSMC. In addition, when tracking the sinusoidal speed reference, suddenly adding rated load to the system and suddenly reducing the rated load mean that the speed change and settling time of the CSMC are 18.1 rpm and 1.59 s, respectively. With the FSMC, the speed change and settling time are reduced to 8.9 rpm and 0.97 s, respectively. Then, by compensating the FSMO to the FSMC, the speed change of the FSMC-FSMO method is further reduced to 6.5 rpm. Therefore, it can be concluded that the FSMC also has faster convergence speed and stronger robustness than the CSMC when tracking the sinusoidal speed reference, and FSMO can also further improve the robustness of the FSMC to the unknown lumped disturbance.

#### 4.1.3. Analysis of Robustness to Mode Uncertainties

According to the analysis in Section 2.1, the dynamical system of the PMSM is also affected by model uncertainties. It can be seen from Equation (Equation 6) that the model uncertainties will cause step or time-varying disturbances to the system. In the above simulation results, the control performance of the proposed method under step disturbance has been verified. Therefore, the simulation results of the three methods under time-varying disturbances are given in this subsection to further verify the robustness to model uncertainties. Then, Figure 6 shows the comparative simulation results of the three methods in tracking step speed signal $\omega_{ref}=100$ rpm under the conditions of adding 2sin(t)+2cos(3t) N·m time-varying load at 5 s. It can be seen that the speed rise of the CSMC at 5 s is 11.2 rpm, and the speed fluctuation of the CSMC caused by the time-varying disturbance is 19.8 rpm. With the FSMC, the speed rise of the CSMC at 5 s and the speed fluctuation of the FSMC are reduced to 5.5 rpm and 5.85 rpm, respectively. From Figure 6d, it can be seen that the FSMO can effectively observe the time-varying disturbance 2sin(t)+2cos(3t) N·m. Then, using the FSMO to compensate the FSMC, the speed rise of the FSMC-FSMO at 5 s and the speed fluctuation of the FSMC-FSMO are further reduced to 3.6 rpm and 0.94 rpm, respectively. Therefore, the proposed FSMC-FSMO method has strong robustness to the disturbance caused by mode uncertainties.

### 4.2. Experimental Results and Analysis

Figure 7 depicts the experimental platform of the PMSM based on the digital control structure of a TMS320F28335 DSP and EP3C40F324 field-programmable array (FPGA), and the corresponding control diagram is illustrated in Figure 8. The DC bus voltage of the inverter is 100 V, and the switching frequency is 10 kHz. The sampling frequency of the current loop is 10 kHz, and the sampling frequency of the speed loop is 1 kHz. The space vector pulse width modulation (SVPWM) and the speed control strategies are carried out in the DSP based on C program. The generation of SVPWM signal, A/D conversion, and encoding signal acquisition are performed on the FPGA. An 18-bit absolute encoder is adopted to detect the position of the rotor. A magnetic powder brake is used to provide external load torque.

#### 4.2.1. Speed Tracking Performance Verification

In order to verify the speed tracking performance of the proposed FSMC-FSMO method, Figure 9 shows the comparative experimental results of speed tracking of the CSMC, FSMC, and FSMC-FSMO methods without load. It can be seen from Figure 9a that the settling time of the CSMC tracking step speed reference is 2.52 s. With the FSMC and FSMC-FSMO methods, the settling time is reduced to 1.11 s. Then, it can be seen from Figure 9b–d that the settling time and speed error fluctuation of the CSMC tracking sinusoidal speed reference are 1.24 s and 4.9 rpm, respectively. With the FSMC, the settling time and speed error fluctuation are reduced to 0.83 s and 1.7 rpm, respectively. Then, the FSMO can further reduce the speed error fluctuation of the FSMC to 1.3 rpm. Therefore, the proposed FSMC has faster speed tracking performance than the CSMC, and the FSMO can further improve the robustness of the FSMC.

#### 4.2.2. Anti-Disturbance Performance Verification

To verify the anti-disturbance performance of the proposed method, Figure 10 shows the comparative experimental results of the CSMC, FSMC, and FSMC-FSMO methods in tracking step speed signal $\omega_{ref}=100$ rpm under the conditions of adding rated load at 10 s and removing rated load at 20 s. Figure 10 shows that the settling time and speed drop of the CSMC when adding rated load at 10 s are 3.20 s and 13.8 rpm, respectively. With the FSMC, the settling time and speed drop are reduced to 1.60 s and 5.4 rpm, respectively. With the FSMC-FSMO, the settling time and speed drop are further reduced to 1.38 s and 3.8 rpm, respectively. When the rated load is removed at 20 s, the settling time and speed rise of the CSMC are 3.70 s and 12.3 rpm, respectively. With the FSMC method, the settling time and speed rise are reduced to 2.28 s and 4.2 rpm, respectively. By compensating the FSMC with the FSMO, the settling time and speed rise are further reduced to 2.12 s and 2.5 rpm, respectively. Therefore, it can be concluded that the proposed FSMC has stronger anti-disturbance than the CSMC, and the FSMO can effectively observe the unknown disturbance and further improve the robustness of the FSMC.

#### 4.2.3. Parameter Robustness Verification

According to the analysis in Section 2.1, the moment of inertia, torque coefficient, and viscous friction coefficient of the PMSM will change with the operating environment and load changes of the motor, causing mode uncertainties. Therefore, it is necessary to verity the parameter robustness of the proposed method. However, it should be noted that the parameters of the PMSM cannot be changed in the practical operation; we change the motor parameters involved in the controller with $J_{0}$ = $J_{0}$→$0.7J_{0}$→$1.3J_{0}$, $K_{t0}$ = $K_{t0}$→$0.7K_{t0}$→$1.3K_{t0}$, $K_{t0}$, and $B_{0}$ = $B_{0}$→$0.7B_{0}$→$1.3B_{0}$. Then, Figure 11 shows the comparative experimental results of the CSMC, FSMC, and FSMC-FSMO methods with parameter change under rated load. It can be seen from Figure 11a that the speed drop of the CSMC, FSMC, and FSMC-FSMO methods is 8.4 rpm, 4.7 rpm, and 4.2 rpm when the moment of inertia $J_{0}$ varies from $J_{0}$ to $0.7J_{0}$, respectively. When the moment of inertia $J_{0}$ varies from $0.7J_{0}$ to $1.3J_{0}$, the speed rise of the three methods is 15 rpm, 8.7 rpm, and 7.6 rpm, respectively. In Figure 11b, the speed drop of the CSMC, FSMC, and FSMC-FSMO schemes is 7.4 rpm, 4.1 rpm, and 3.6 rpm when the torque coefficient $K_{t0}$ varies from $K_{t0}$ to $0.7K_{t0}$, respectively. When the torque coefficient $K_{t0}$ varies from $0.7K_{t0}$ to $1.3K_{t0}$, the speed rise of the three methods is 12.3 rpm, 7.3 rpm, and 5.9 rpm, respectively. Finally, it can be seen from Figure 11c that the change in the viscous friction coefficient $B_{0}$ has little impact on the speed control performance. Therefore, it can be concluded that the FSMC has stronger parameter robustness than the CSMC, and the FSMO can further enhance the parameter robustness of the FSMC method.

## 5. Conclusions

This paper has proposed a fast and robust FSMC-FSMO scheme for PMSM speed regulation using the fixed-time control theory. Targeting the slow convergence speed and poor robustness of the CSMC, an FSMC was proposed using the fixed-time control theory to achieve rapid convergence and strong anti-disturbance performance of the PMSM system. Then, the FSMO is applied as a compensator to further improve the robustness of the FSMC and attenuate the sliding mode chattering in the speed tracking. Finally, the stability and fixed-time convergence property of the proposed method are proofed by the Lyapunov method. Simulation and experimental results show that the proposed control method can obtain faster response speed and stronger robustness than the CSMC scheme. Meanwhile, the fixed-time convergence property of the FSMC is verified by the comparative simulation results of the CSMC and FSMC under different initial speed references. In addition, this method can also be used for current control of the PMSM, as well as other types of motors, such as induction motors, linear motors, reluctance motors, etc.

## Figures and Tables

**Figure 1 sensors-24-01561-f001:**
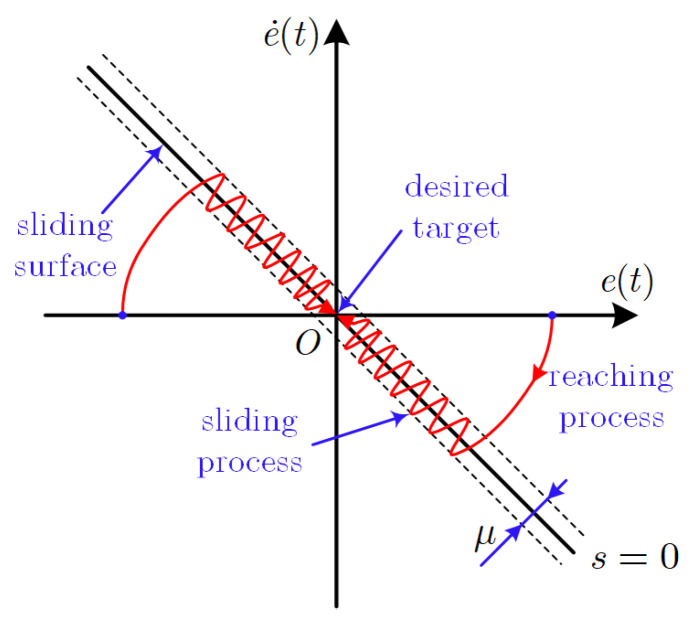
The basic sliding motion of SMC.

**Figure 2 sensors-24-01561-f002:**
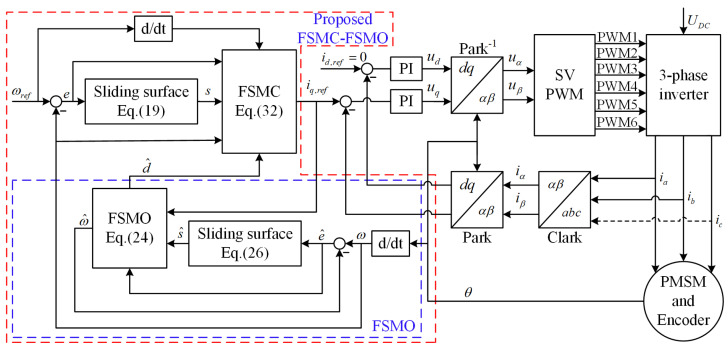
Structural diagram of PMSM drive system based on FSMC-FSMO scheme.

**Figure 3 sensors-24-01561-f003:**
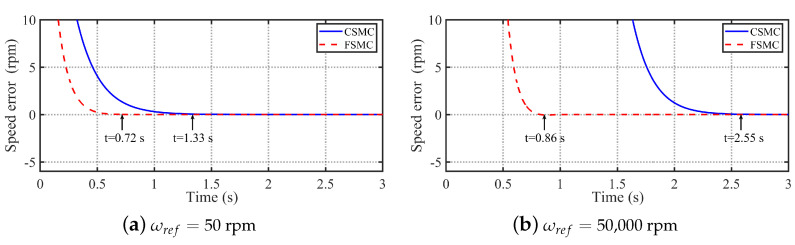
Comparative simulation results of the CSMC and FSMC methods under different initial conditions when d=0. (**a**) Simulation results with $\omega_{ref}=50$ rpm. (**b**) Simulation results with $\omega_{ref}=50,000$ rpm.

**Figure 4 sensors-24-01561-f004:**
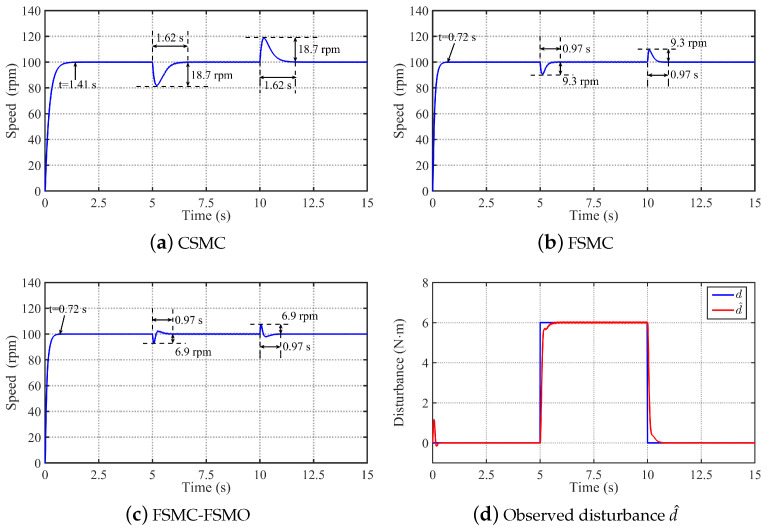
Comparative simulation results of the CSMC, FSMC, and FSMC-FSMO methods in tracking step speed signal $\omega_{ref}=100$ rpm under the conditions of adding rated load at 5 s and removing rated load at 10 s. (**a**) Simulation result with the CSMC. (**b**) Simulation result with the FSMC. (**c**) Simulation result with the FSMC-FSMO. (**d**) Observed disturbance $\hat{d}$ by the FSMO.

**Figure 5 sensors-24-01561-f005:**
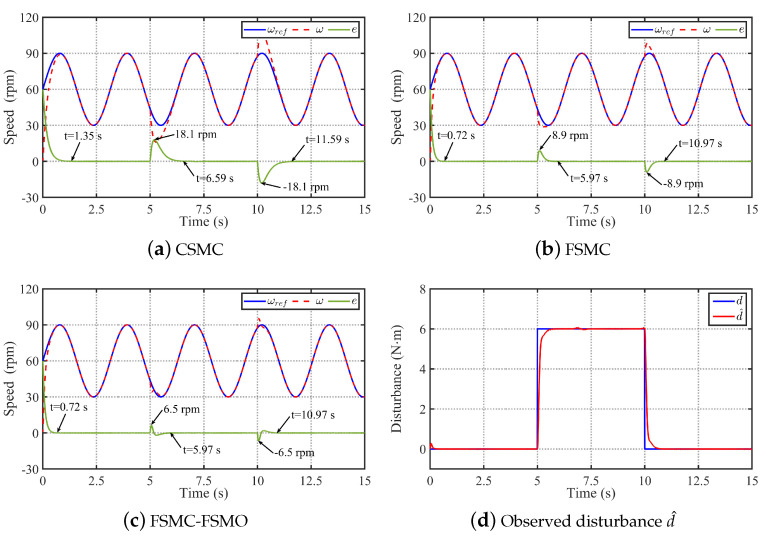
Comparative simulation results of the CSMC, FSMC, and FSMC-FSMO methods in tracking sinusoidal speed signal $\omega_{ref}=60+30sin\left(2t\right)$ rpm under the conditions of adding rated load at 5 s and removing rated load at 10 s. (**a**) Simulation result with the CSMC. (**b**) Simulation result with the FSMC. (**c**) Simulation result with the FSMC-FSMO. (**d**) Observed disturbance $\hat{d}$ by the FSMO.

**Figure 6 sensors-24-01561-f006:**
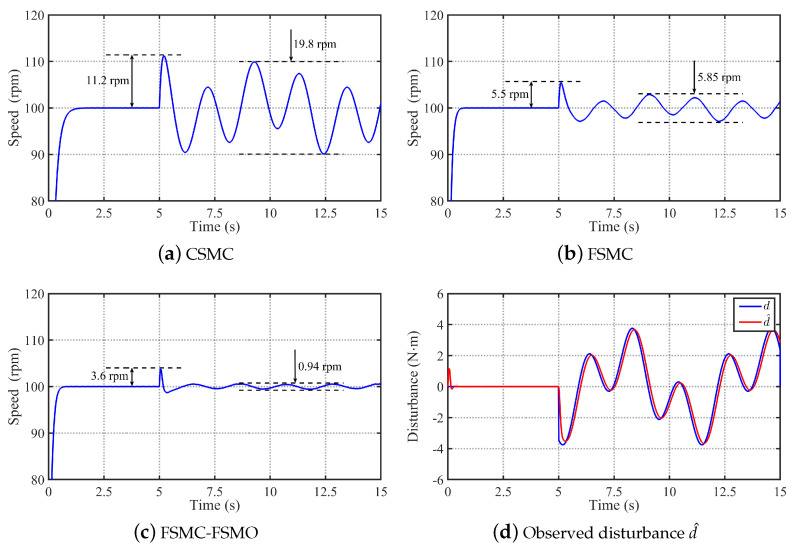
Comparative simulation results of the CSMC, FSMC, and FSMC-FSMO methods in tracking step speed signal $\omega_{ref}=100$ rpm under the conditions of adding 2sin(t)+2cos(3t) N·m time-varying load at 5 s. (**a**) Simulation result with the CSMC. (**b**) Simulation result with the FSMC. (**c**) Simulation result with the FSMC-FSMO. (**d**) Observed disturbance $\hat{d}$ by the FSMO.

**Figure 7 sensors-24-01561-f007:**
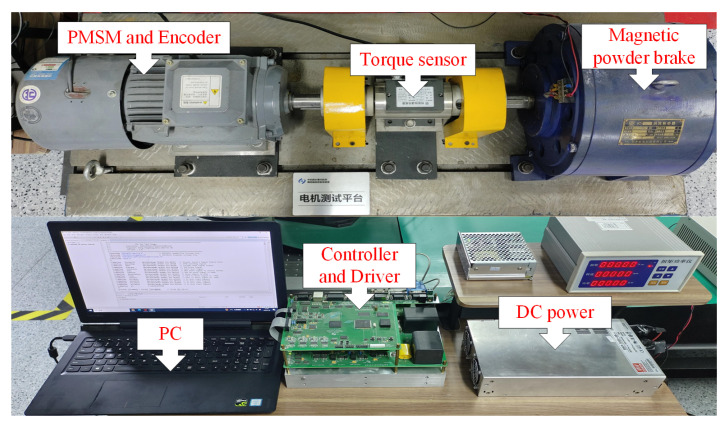
Photograph of the experimental platform of the PMSM.

**Figure 8 sensors-24-01561-f008:**
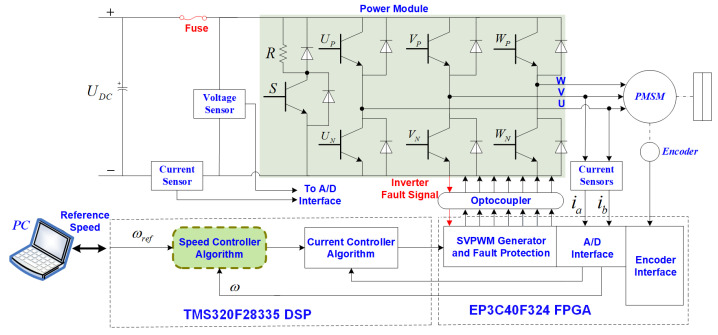
Structural diagram of the PMSM servo system.

**Figure 9 sensors-24-01561-f009:**
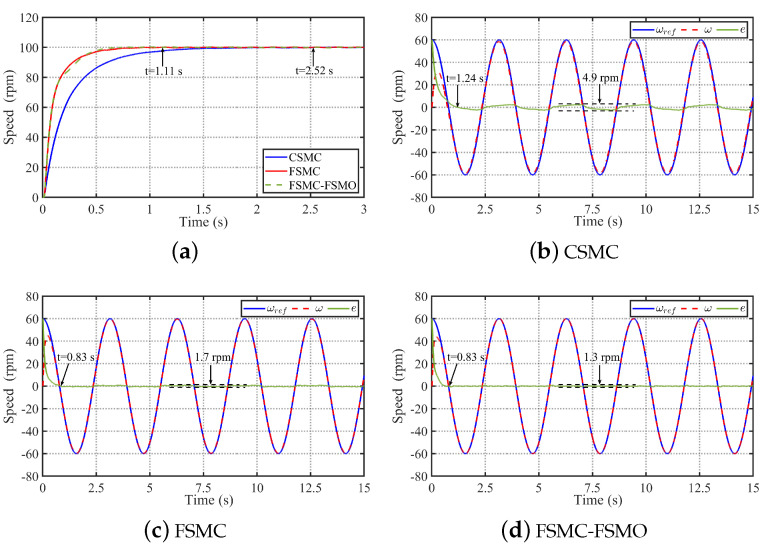
Comparative experimental results of speed tracking of the CSMC, FSMC, and FSMC-FSMO methods without load. (**a**) Experimental result in tracking step speed signal $\omega_{ref}=100$ rpm. (**b**) Experimental result of the CSMC in tracking sinusoidal speed signal $\omega_{ref}=60cos\left(2t\right)$ rpm. (**c**) Experimental result of the FSMC in tracking sinusoidal speed signal $\omega_{ref}=60cos\left(2t\right)$ rpm. (**d**) Experimental result of the FSMC-FSMO in tracking sinusoidal speed signal $\omega_{ref}=60cos\left(2t\right)$ rpm.

**Figure 10 sensors-24-01561-f010:**
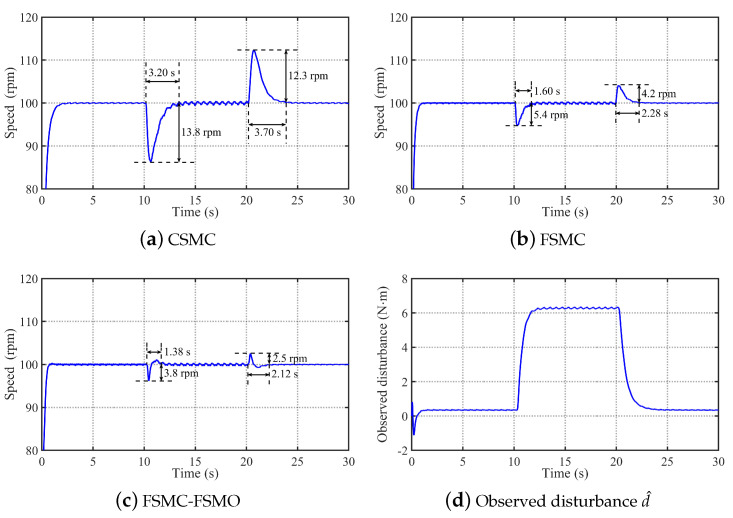
Comparative experimental results of the CSMC, FSMC, and FSMC-FSMO methods in tracking step speed signal $\omega_{ref}=100$ rpm under the conditions of adding rated load at 10 s and removing rated load at 20 s. (**a**) Experimental result with the CSMC. (**b**) Experimental result with the FSMC. (**c**) Experimental result with the FSMC-FSMO. (**d**) Observed disturbance $\hat{d}$ by the FSMO.

**Figure 11 sensors-24-01561-f011:**
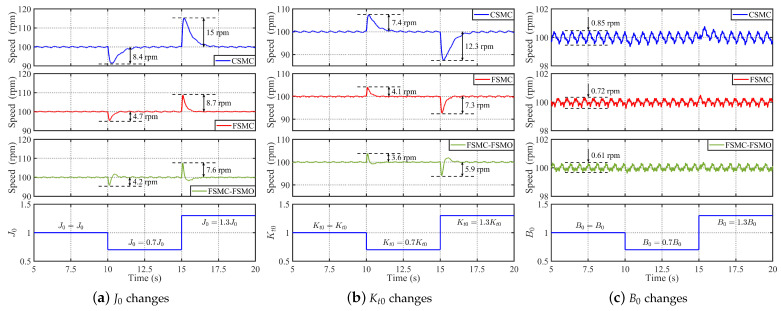
Comparative experimental results of the CSMC, FSMC. and FSMC-FSMO methods with parameter change under rated load. (**a**) Experimental result with $J_{0}$ changes. (**b**) Experimental result with $K_{t0}$ changes. (**c**) Experimental result with $B_{0}$ changes.

**Table 1 sensors-24-01561-t001:** The nominal parameters of the PMSM.

Symbols	Characteristics	Values
$U_{DC}$	DC-bus-rated voltage	100 V
$I_{N}$	Rated current	5 A
$T_{l}$	Rated torque	6 N·m
$n_{p}$	Number of pole pairs	3
$\psi_{f}$	Rotor flux linkage	0.29 Wb
$R_{s}$	Stator resistance	0.675 Ω
$L_{s}$	dq-axis inductance	0.0065 H

## Data Availability

The data are contained within the article.

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
