# Peer review of "Fixed-Time-Convergent Sliding Mode Control with Sliding Mode Observer for PMSM Speed Regulation"

_sensors, 2024, doi:10.3390/s24051561_

Round 1
Reviewer 1 Report
Comments and Suggestions for Authors
This paper considers a fast and robust FSMC-FSMO scheme for PMSM speed regulation using the fixed time control theory for improving the convergence speed. The simulation and experimental results show the effectiveness of the proposed paper. Generally speaking, the topic is interesting and organized well, I have some minor comments:
1. Why the uncertainty d in line 193 can be treated as a constant, is this the case in practical situations?
2. The motivation should be stated more clearly in Introduction.
3. Some new literature regarding SMO are missed in the Introduction, such as
10.1109/TAC.2017.2665699;
10.1109/TSMC.2020.3034746;
10.1016/j.automatica.2020.109274;
10.1016/j.automatica.2022.110676
4. Since the SMC and SMO can be convergent independent on the initial conditions, the authors should test this point in simulation part.
The reviewer suggest authors to consider the above comments before the acceptance.
Reviewer 2 Report
Comments and Suggestions for Authors
In this article has proposed a fast and robust FSMC-FSMO scheme for PMSM speed regulation using the fixed time control theory. The FSMC was proposed using the fixed-time control theory to achieve rapid convergence and strong anti-disturbance performance of the PMSM system. The FSMO is applied as a compensator to further improve the robustness of the FSMC and attenuate the sliding mode chattering in the speed tracking. The stability and fixed-time convergence property of the proposed method are proofed by Lyapunov method.
Simulation and experimental results shows that the proposed control method can obtain faster response speed and stronger robustness than the CSMC scheme.
Question / reminder:
- row 280 and Figure 3(b) - why the reference speed is till 50 000 rpm?
For PMSM with nominal parameters shown in Table 1 is it a very large value (unreal) - explain it in the article.
Round 2
Reviewer 1 Report
Comments and Suggestions for Authors
The revision is satisfactory. It can be accepted now.